# Single-Cell Approaches to Deconvolute the Development of HSCs

**DOI:** 10.3390/cells10112876

**Published:** 2021-10-25

**Authors:** Yang Xiang, Ryohichi Sugimura

**Affiliations:** Li Ka Shing Faculty of Medicine, School of Biomedical Sciences, University of Hong Kong, Hong Kong SAR, China; xiangyang044@gmail.com

**Keywords:** HSC, single-cell technology, molecular barcode, spatial transcriptomics

## Abstract

Hematopoietic stem cells (HSCs) play a core role in blood development. The ability to efficiently produce HSCs from various pluripotent stem cell sources is the Holy Grail in the hematology field. However, in vitro or in vivo HSC production remains low, which may be attributable to the lack of understanding of hematopoiesis. Here, we review the recent progress in this area and introduce advanced technologies, such as single-cell RNA-seq, spatial transcriptomics, and molecular barcoding, which may help to acquire missing information about HSC generation. We finally discuss unresolved questions, the answers to which may be conducive to HSC production, providing a promising path toward HSC-based immunotherapies.

## 1. Efforts on HSC Identification and Generation

Hematopoietic stem cells (HSCs) are crucial for the maintenance and production of blood cells. During murine embryogenesis, HSCs originate from the aorta-gonad-mesonephros (AGM) at embryonic day 10.5 (E10.5), then circulate through the fetal liver and spleen, and finally reside in the bone, wherein they maintain themselves [1]. However, despite research efforts in the last few decades, the markers used to define HSCs, the method by which HSCs can be directly produced, and the way in which HSCs maintain themselves remain unclear. Early research on HSCs dates back to the 1960s when Till and McCulloch exploited Colony-Forming Unit Spleen Methods (CFU-spleen) and proved the self-renewal and differentiation of bone marrow-derived stem cells, which are now known to be multipotent progenitors [2]. Later, Irving L. Weissman and colleagues determined that the Sca1^+^ c-kit^+^ CD41^+^ population contains HSCs [3]. Then, Sean J. Morrison identified SLAM family proteins as markers, and then engraftment experiments confirmed that the CD150^+^ CD48^−^ cell population contains functional HSCs [4]. However, understanding how these HSCs directly emerge is a challenge. Many researchers have proposed markers to define direct precursors of HSCs (pre-HSCs): for example, mesenchyme CD45^−^ CD31^−^ VEcad^−^ [5], subaortic patches CD45^−^ cKit^+^ AA4.1^+^ [6], and hemogenic endothelial cell CD45^−^ VEcad^+^ [7]. Subsequently, Jean-Charles Boisset used in vivo time-lapse confocal imaging to confirm that hemogenic endothelial cells are precursors of HSCs. Upon their emergence at AGM and maturation in the fetal liver, HSCs migrate to the bone marrow, where they are maintained. At present, the best-described niche for HSC maintenance in the bone marrow is the perivascular HSC niche [1]. Within the niche, stromal cells (CXCL12^+^ reticular cells), mesenchymal stem cells (CXCL12^+^ SCF^+^), and endothelial cells (gp130 cytokine receptor^+^ VEGFR2^+^) around blood vessels contribute to the maintenance of HSCs. Several other factors are proposed to maintain HSCs to a certain extent. Shahin Rafii and colleagues demonstrated that endothelial cells support the self-renewal and repopulation of hematopoietic stem cells based on Notch signaling [8]. Linheng Li and colleagues illustrated the role of quiescent Flamingo Fmi and Frizzled8 in the niche [9]. Lei Ding and colleagues discovered that hepatic thrombopoietin, a systemic factor restrictedly expressed by hepatocytes, sustains HSCs [10]. In this review, we mainly focus on the development of HSCs (hemogenic process) and the potential exploitation of advanced technologies, such as molecular barcoding, single-cell RNA-seq, and spatial transcriptomics, to elucidate the missing steps during HSC generation.

### 1.1. Attempts to Generate HSCs 

In the past decade, many have attempted to understand how HSCs develop in vivo. A unique sequence of Hedgehog/Notch/Scl signaling generates hemogenic endothelium in the mouse via endothelial-to-hematopoietic transition (EHT) [11]. In vivo imaging revealed that human ESC-derived HE undergoes an EHT similar to the mouse HE [12]. The molecular characterization of HE and the subsequent process of EHT will be crucial for the generation of HSCs from human PSCs [13]. To produce engraftable HSCs as curative sources for various hematological malignancies and immunodeficiency, understanding the development of HSCs on the basis of research and clinical perspectives is crucial [14]. Efforts to generate HSCs have been extensive. Gordon Keller and colleagues proved that CD34^+^ VEcad^+^ progenitor cells with characteristics of hemogenic endothelium can produce myeloid and erythroid cells [15]. Later, they generated HE from hPSC in vitro and proved that the HE lineage based on Notch signaling is restricted to the CD34^+^ CD43^−^ CD184^−^ CD73^−^ DLL4^−^ CD45^−^ population (Figure 1) [13]. Furthermore, researchers have succeeded in producing HSCs in vivo. Daniel G. Tenen and colleagues confirmed that when reprogrammed iPS cells are injected into immune-deficient NSG mice, they can generate HSCs through teratoma formation (Figure 1) [16]. In addition, Hiromitsu Nakauchi and colleagues further proved that iPS-derived HSCs generated from teratoma could reconstitute the hemato-lymphopoietic system in second recipients [17]. In conclusion, although PSCs can generate engraftable HSCs in teratoma, HE cannot engraft. This indicates that certain factors remain unknown. Before moving to the next section, which discusses ways to identify missing links, we first refer to two practical methods to generate HSCs in vitro and in vivo in current use: (1) directed differentiation strategies by cell-extrinsic cytokines/morphogens or (2) direct conversion through transcription factors (TFs).

Here, we focus on the second strategy. TFs are overexpressed in either (1) PSCs for forward programming into a particular lineage or (2) mature cell types for direct conversion into less committed precursors. Igor Slukvin and colleagues screened 27 candidate genes and revealed that TFs could convert hPSCs into hemogenic endothelium with myeloid or erythro-megakaryocytic potential; for example, ETV2 and GATA2 induce hematoendothelial with pan-myeloid potential, and GATA2 and TAL1 generate hemato-endothelial with erythro-megakaryocytic potential [18]. George Q. Daley and colleagues screened a mini-library of transcription factors and identified that HOXA9, ERG, and RORA (EAR) were essential for the activation of the self-renewal program in hPSC-derived progenitors with erythroid potential, but they failed to generate HSCs with long-term and multilineage repopulation potential [14]. In other words, no functional HSC-like cells emerged. In 2014, Derrick J. Rossi and colleagues produced induced HSCs (iHSCs) from mouse B progenitor cells, demonstrating that iHSCs are capable of multilineage differentiation potential, reconstituting stem/progenitor compartments, and are practicably transplantable with the same robustness as genuine HSCs [19]. However, the same strategy was not adopted in human B progenitor cells. In another study, George Q. Daley and colleagues conducted in vivo screening of TFs and identified seven transcription factors (ERG, HOXA5, HOXA9, HOXA10, LCOR, RUNX1, and SPI1) that enable the development of engraftable HSC-like cells from hPSCs. The seven factors ensure that derived HSC-like cells are endowed with multilineage potential in primary and secondary mouse recipients [20]. However, the production of HSC-like cells remains low and depends on in vivo conditions. Thus, the in vitro generation of functional HSCs and their derivative immune cells will be the next milestone in the field. 

### 1.2. Single-Cell RNA-Seq Deconvolution for Missing Links in HSC Generation

Although the markers of HSCs and their progenitors have provided a snapshot image of HSC development, the dynamic trajectory from hemogenic endothelial cells to HSCs is still unclear. Based on recent research articles, there are several types of missing links [21,22,23]:
The heterogeneity of the precursors pool;The developmental trajectory and molecular changes for HSC themselves (such as transcriptome and epigenetic information);The microenvironmental support in vitro and in vivo.

Recent advances in single-cell RNA-seq provide great insights into these issues. Single-cell RNA-seq resolves the complex heterogeneity of the precursors pool, depicting the hierarchical relationship among subpopulations. Transcriptome mapping between in vitro hPSC-derived precursors and their in vivo counterparts is expected to decipher missing environmental links in HSC generation [24,25].

As for the complexity of the precursors pool, single-cell RNA-seq has validated the arterial origin of HSCs, confirming that CD44^+^ arterial endothelial cells have the hemogenic potential [26].

As to molecular changes during HSC production, researchers have applied pseudotime analysis based on the transcriptome profile to analyze dynamic gene expression during fate changes between primitive and definitive hematopoiesis. They found that ROCK inhibition in humans promotes definitive hematopoiesis that generates multipotent progenitors (MPPs) in hPSC differentiation [26]. Another study demonstrated that long non-coding RNAs play a role in HSC generation comparable to that of mRNAs. The study demonstrated that lncRNA H19 deficiency leads to impaired HSC generation via hypermethylated promoters and the inhibited expression of TFs such as Runx1 and Spi1 [27].

Lastly, single-cell RNA-seq revealed the difference between in vivo and in vitro pre-HSCs in a cultured microenvironment. Irwin Bernstein and colleagues identified that Notch receptors, VLA-4 integrin, and CXCR4 were sufficient to support the generation of functional HSCs from HE in vitro [28].

To conclude, single-cell technology can decipher the heterogeneity of pre-HSCs and help to discover rare populations (CD44^+^ arterial endothelial cells) that can differentiate into HSCs. Considering the rarity of pre-HSCs that ultimately contribute to HSCs, the current methods of bulk-level analysis (RNA-seq, ATAC-seq, and ChIP-seq) cannot capture the needle in the haystack. Single-cell transcriptomic data provide deep insight into the types of factors that are potential driving forces of HSC development. The highlighted transcription factors (Runx1 and Spil1) or cell surface proteins (Notch receptors, VLA-4 integrin, and CXCR4) are conducive to the generation of HSCs, but the deterministic factors are yet to be identified. These factors arise as the usual suspects suggested by prevailing knowledge from murine models and zebrafish models. We assume the unique factors in human hematopoiesis that single-cell technologies will capture. According to George Q. Daley and colleagues’ work [20], to construct a minimized factor pool, the first step is to knock in and overexpress these factors in PSCs in vitro. Secondly, PSCs are driven to differentiate into HSCs by morphogen/cytokine-based methods, transcription factor-based strategies, or a combination of these approaches either in vitro or in vivo. Finally, high production of HSCs can be anticipated. The upcoming findings of HSC development revealed by single-cell technology will accelerate the discovery of new factors and the application of these factors to HSC generation (Figure 2).

## 2. New Single-Cell Techniques Address HSC Generation at Spatial and Temporal Resolutions

Single-cell RNA-seq entails the analysis of single cells outside of their native spatial context. Much information about the cell environments and locations is lost. Moreover, single-cell RNA-seq, incapable of capturing the original spatial information, fails to relate the cellular context to gene expression [29]. Recent advances in spatial omics methods and molecular barcode strategies may provide alternative solutions to these problems.

### 2.1. Spatial Single-Cell RNA-Seq 

Xiaowei Zhuang and colleagues developed a new spatial single-cell RNA-seq strategy termed ‘MERFISH’. Each RNA species bound four encoding probes of N probes. During the sequencing process, if fluorescence was detected in one RNA species, then that RNA received “1” bit. Each RNA with four “1”s could then be identified. The researchers detected 1000 transcripts from 100 cells within 16 rounds, and they continued to make improvements in the detected numbers of RNA transcripts and the number of cells analyzed at the same time. Recently, Siyuan Wang and his colleagues applied Merfish to the investigation of the spatial transcriptomic profile of single cells in fetal livers of Ter2^−/−^ and WT mice. They found that the loss of Ter2 augmented the HSC population and enhanced Wnt and Notch signaling genes in the HSC niche. They also validated that HSCs directly interacted with endothelial cells (ECs), and they showed that two subtypes of ECs largely made different contributions to the HSC niche via their unique signaling molecules. Finally, hepatocytes and megakaryocytes were discovered to support HSC niches [30,31].

Aviv Regev, Alexander F Schier, and colleagues used RNA in situ to generate a binary spatial reference map that reflects the detection and expression of each landmark gene within each spatial domain. They designed a novel bioinformatics package called Seurat. The algorithm used single-cell RNA profiles in sequenced cells to input the expression of each landmark gene in each cell. Then, it related the imputed RNA expression levels to the binary spatial expression value in the reference map. Finally, it inferred the spatial origins of a sequenced cell by posterior probability [29].

However, serial RNA in situ methods work with targeted and pre-selected genes. To lower labor costs and discover unknown expressed genes, more convenient methods are needed. In 2016, Joakim Lundeberg and colleagues devised a strategy called Spatial Transcriptomics [32]. First, they introduced positioned reverse transcription oligo(dT) with molecular barcodes on glass slides and then placed fresh tissue on the slides. After permeabilization, they added reverse transcription reagents and used fluorescence-labeled nucleotides to visualize cDNA. The results showed that the pattern of fluorescent cDNA was similar to the tissue structure observed using immunohistology. 

Subsequently, in 2019, Slide-seq was developed by Fei Chen, Evan Z. Macosko, and colleagues, and its resolution was higher than that of spatial transcriptomics [33]. Slide-seq was performed as follows (Figure 3a). Firstly, researchers packed distinct DNA-barcoded microbeads onto a coverslip and then determined each bead’s barcode sequence by SOLID (sequencing by oligonucleotide ligation and detection). Subsequently, fresh-frozen tissue sections were transferred onto the dried bead surface, and mRNA from the tissue was captured by beads. Then, they digested the tissue part, released the beads, harvested mRNA, amplified the library, and prepared RNA for sequencing. However, there are two major obstacles to the application of Slide-seq. First, the 100 μm array spots in spatial transcriptomics are not small enough or densely packed enough to achieve single-cell resolution. Second, the transcript detection sensitivity is relatively low; for example, each bead captured no more than 100 transcripts [34,35]. Furthermore, few Slide-seq studies have been reported in the field of HSC generation.

Although there is a diverse set of spatial transcriptomic methods, some require high-cost techniques and bespoke equipment. Thus, an available commercial kit has been in high demand [36]. For instance, in the commercialized 10× Visium platform, sliced tissue sections are placed onto the commercial slides and permeabilized, and mRNAs are captured by primers on the spots and then converted into cDNA by reverse transcription. The cDNAs generated within one spot share a common barcode. Finally, gene expression can be mapped with the acquired initial tissue images [37]. The 10× Visium platform enables high-throughput and large-scale studies of gene expression within a spatial context. This can advance many research fields, such as neuroscience, immunology, hematology, and gastroenterology.

In summary, spatial transcriptomics could reveal the environment and location of cells. Undoubtedly, some new regulatory factors can be uncovered. Such findings would be crucial to the development of HSCs. PSCs should be endowed with factors by knock-in approaches and preferably induced into HSCs. In addition, new microenvironmental factors such as cytokines can be engineered in PSCs to further support HSC generation.

### 2.2. Molecular Barcoding

Although recent spatial transcriptomic methods allow genome-scale measurements, such experiments are usually restricted to limited tissue regions and unable to detect every cell in a specific organ. Recently, molecular barcoding methods have paved the way for the mentioned issues (Figure 3b). Before single-cell RNA-seq was invented, people traditionally used lentiviral static barcodes in HSCs. For instance, Irving L. Weissman and colleagues designed a static barcode containing a common library ID with a 6-bp sequence and a unique cellular label with a random 27-bp sequence. Then, they transduced barcodes into HSCs, confirmed barcode recovery and diversity, and found different differentiation patterns within HSC clones through a combination of DNA barcoding and high-throughput sequencing (HTS) [38]. In the following years, Connie J. Eaves and colleagues tracked the in vivo growth and differentiation of human CD34^+^ cord blood cells in secondary NSG mice recipients by static lentivirus barcodes [39]. In addition to static barcodes, there is another type of barcoding method: evolving barcodes [40]. For cumulative barcoding in cell dynamic evolution, when barcoded parental cells divide, their barcodes will be edited, and daughter cells will be endowed with the newly edited barcodes. Thus, cumulative barcoding is instructive in cell fate-tracing. Recent research papers mainly focus on the following several molecular barcoding strategies.
Transposon (Transposase-mediated random integration barcoding);CARLIN (CRISPR array repair lineage tracing);LARRY (lineage and RNA recovery);PolyloxExpress (Cre-recombinase-dependent barcoding);GESTALT (genome editing of synthetic target arrays for lineage tracing).

A series of works by Fernando D. Camargo and colleagues made a pioneering contribution to redefining the populations that play a major role in native or transplantable hematopoiesis. They designed a novel barcode system called inducible transposon tagging. By labeling the progenitors and their progenies, they surprisingly revealed that LT-HSCs predominantly sustain transplantable hematopoiesis, whereas multipotent MPPs maintain native hematopoiesis [41]. Another system, CARLIN, demonstrated the lineage trajectory and cellular phenotyping at a single-cell level. The researchers validated that this system has high editing efficiency and produces a large number of barcodes in the mouse ESC in vitro and in vivo; then, they reconstructed the edited ESC lineage, resolved the controversy on how much the individual HSC contributes to regeneration, and identified the heterogeneity and active cell states of HSCs [42]. They established a LARRY lentiviral barcoding library. This static barcode system was used to mark long-term hematopoietic stem cells and analyze the low progeny output activity of HSCs, indicating limited lineage commitment. They found that targeting TCF-15 unlocked the potential of HSCs [43].

In 2020, Hans-Reimer Rodewald and colleagues designed the PolyloxExpress system, which makes use of the Cre-recombinase-dependent molecular barcode, and defined three groups of HSC clones: differentiation-inactive HSCs, myelo-erythroid biased, and multilineage HSCs. They proposed that this new hierarchy might be suitable, because currently defined dormant cells are not differentiation-inactive HSCs, but have characteristics of multilineage HSCs [44].

In 2021, Christopher J Lengner and colleagues made improvements in GESTALT and developed macsGESTALT (multiplexed, activatable, clonal, and subclonal GESTALT), a new system with inducible Cas9, independent gRNA, and barcode construct. Specifically, the barcode vector contains evolving barcode regions and static barcode parts. The researchers confirmed the practicality of macsGESTALT in vitro with limited dilution and expansion and then, proved its ability to reconstruct colonies and identify the EMT continuum in PDAC-bearing mice [45]. However, this technique has not been applied to understanding the development of HSCs.

In 2021, somatic mutations are considered DNA barcodes and differentiate cell lineages, because lineages develop as a result of asymmetrical mitosis. Christopher J Lengner, Flora M. Vaccarino, and colleagues demonstrated that DNA repair efficiency could be attributed to this lineage imbalance. Based on the criteria that variants are shared by cell clones, and that in a clone, variants from continuous cell divisions should gradually decrease, the researchers built the branch-based lineage and found that a large imbalance occurred for the first two blastomeres, a dominant and a recessive cell lineage. Notably, a high fraction of indels occurred among the recessive lineage compared with the dominant lineage, which indicated that DNA repair efficiency contributed to this imbalanced lineage differentiation. Further analysis confirmed that most SNVs resulted from the consecutive deamination of 5-methylcytosine [46]. Another paper in the same issue of *Science* led by Peter J. Park, Christopher A. Walsh, and colleagues revealed a similar mechanism. The researchers performed high-depth whole-genome sequencing (WGS) in different human tissues to identify large numbers of somatic single-nucleotide variants (sSNVs). Based on the use of these variants as barcodes, both studies reconstructed an early embryonic development lineage and revealed that there was an asymmetric partitioning of early progenitors, and a subset of effective progenitor pools made contributions during blastula formation, gastrulation, and organogenesis [47]. 

These molecular barcode strategies have unique internal principles and respective limitations. CARLIN, Macs-GESTALT, Polylox, and Transposon are evolving barcode methods, whereas LARRY is a type of static barcode (Table 1). CARLIN and Macs-GESTALT, which are based on the CRISPR-Cas9 editing system, may face problems of barcode loss and a low detection rate of transcribed barcode RNAs. Polylox is relatively complex and costly because PacBio Single-Molecule Real-Time (SMRT) Sequencing is needed. Transposon barcodes may be labor-intensive because they require a large set of PCR primers, and they could lead to large fragment loss or unknown genome alterations in divided cells because of random integration. Somatic mutations as a new type of barcode could be used as an indicator of non-invasive lineage tracing and fate mapping in human tissues and cells without introducing exogenous cassettes into cell genomes. Although molecular barcode strategies are not sufficiently mature, we predict that ongoing efforts will increase the detection rate of barcode transcripts and decrease damage to cells. Soon, we could identify the sets of expressed or silenced genes that control the development of HSCs by comparing HSCs with precursor cells (pre-HSCs) by giving each progenitor cell and its progeny a unique identity simultaneously. We hope that these strategies will be applied frequently and advance the understanding and production of HSCs.

## 3. Outlook for HSC Generation

Based on our knowledge of hematopoiesis, there are several key nodes in the development of HSCs, such as endothelial cells (ECs), hemogenic endothelial cells (HECs), and hematopoietic stem cells (HSCs). However, the production of HSCs is not a clear-cut linear process. Many factors are involved in the development from source cells to HSCs. For instance, different combinations of transcription factors (TFs) and heterogenous HSC niche cell supporters suggest that more than one program drives the fate of HSCs. At present, the combinations of factors that directly drive the decision to develop into HSCs remain unclear [48].

We predict that some questions will be resolved by recent advanced techniques, such as molecular barcoding and spatial transcriptomics combined with single-cell RNA-seq. First, barcode-related lineage tracing bridges the connection between progenitor cells and progenies, not only revealing the information of major clusters, but also dissecting the origins and distributions of subclones. Second, cell–cell interactions can be mapped within their spatial context by spatial transcriptomics. Finally, as we continue to uncover the panorama of the development of HSCs, we could integrate previously collected data and predict the best path for HSC generation based on Bayesian models. We are tantalizingly close to the ability to efficiently produce HSCs due to the guidance and multiple combinations of advanced methods from mathematics, bioinformatics, and stem cell biology.

## Figures and Tables

**Figure 1 cells-10-02876-f001:**
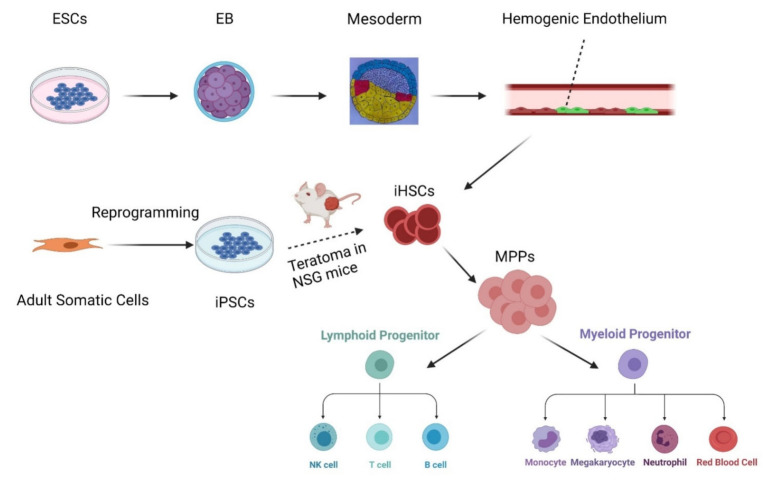
HSCs emerge either via embryo-derived HE or via iPSCs. Induced HSCs can differentiate into MPPs and then into lymphoid and myeloid progenitors, which finally maturate into NK/T/B cells or monocytes/megakaryocytes/neutrophils/red blood cells, respectively. ESCs: embryonic stem cells; EB: embryoid body; HE: hemogenic endothelium; iPSCs: induced pluripotent stem cells; iHSCs: induced hematopoietic stem cells; MPPs: multipotent progenitors. (The figure was created with BioRender.com (accessed on 24 October 2021)).

**Figure 2 cells-10-02876-f002:**
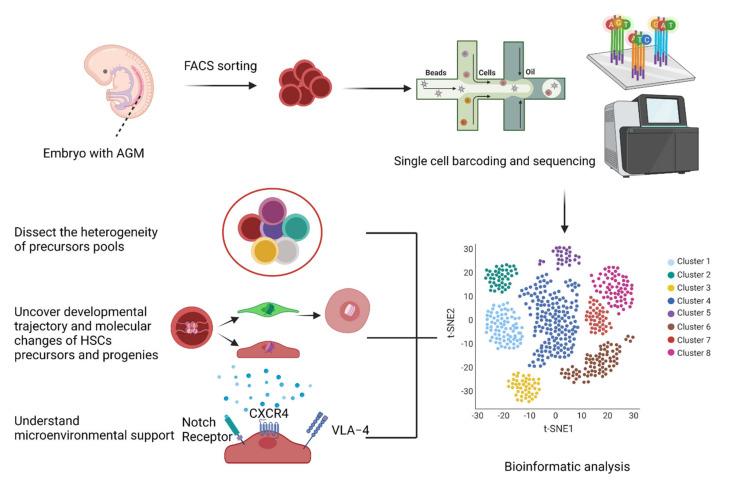
Schematic overview of the application of single-cell RNA-seq in deciphering the complexity of precursors pools, developmental trajectory or molecular changes in HSC pools, and environmental supporting signals. Both the time point of the embryos (E10.5 in mice) and the resolution of transcriptomics hold the key to identifying the crucial factors that define HSCs, such as Notch, CXCR4, and VLA-4. (The figure was created with BioRender.com (accessed on 24 October2021)).

**Figure 3 cells-10-02876-f003:**
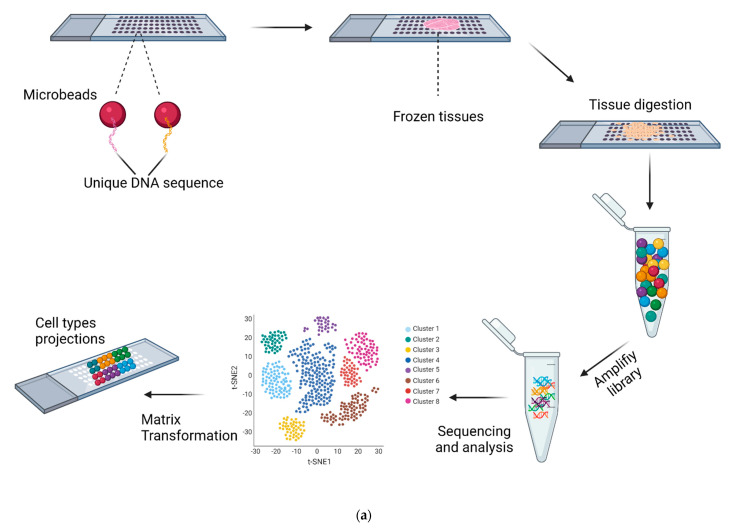
(**a**) Schematic representation of Slide-seq. Microbeads in each well decorated with a unique DNA sequence are deposited. Samples are placed on the slide, and reverse transcription follows. Then, the library is amplified, and single-cell RNA-sequencing is performed. In the end, projected cell types in the microwell are determined by matrix transformation and assignment of the sequenced datasets to Slide-seq beads. (The figure was created with BioRender.com (accessed on 24 October 2021)). (**b**) DNA static barcode lacks the resolution of cell heterogeneity (**left**). In contrast, evolving DNA barcodes provide a higher resolution of cell heterogeneity (**right**). Barcodes repeatedly label a dividing cell clone, giving every progeny a distinct identity, in theory, at sequential levels of lineage hierarchy. (The figure was created with BioRender.com (accessed on 24 October 2021)).

**Table 1 cells-10-02876-t001:** Molecular Barcoding Strategies. The tables summarize recently established evolving barcodes. Once barcodes are transcribed as mRNAs, single-cell RNA-seq could deconvolute both lineage trajectory and cell type taxonomy.

Technique	Barcode Type	Static Barcodes or Evolving Barcodes?	Barcode as mRNA?
CARLIN	INDEL	Evolving barcodes	Yes
Macs-GESTALT	INDEL	Static barcodes & Evolving barcodes	Yes
Poly-Lox Express	Recombination	Evolving barcodes	Yes
LARRY	Integration	Static barcodes	Yes
Transposon	Random Integration	Evolving barcodes	No

Note: Indel: insert or deletion.

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
