# Peer review of "Single-Cell Approaches to Deconvolute the Development of HSCs"

_cells, 2021, doi:10.3390/cells10112876_

Round 1

Reviewer 1 Report

In this review, the recent progress of PSC and advanced technologies, such as molecular barcoding, single-cell RNA-seq, spatial transcriptomics, and organoid-related platforms are summarized. The review for the application of single cell technology to understand the production of HSCs and developmental hematopoiesis from PSCs will benefit the community. The biggest problem of this review is that, there is no enough description on how “single cell” helps “hematopoietic cells from PSCs” research. The manuscript lacks the summary on the achieved findings on PSCs to hematopoiesis by single cell technology, also lacks the perspective of single-cell on PSCs research.

Section 2 describes the schematic overview of single cell in deciphering Hematopoietic Stem and Progenitor Cells. It is good to summarize some the technical development, but the review does not describe how these development help understanding hematopoiesis from PSCs. Same situation is for spatial single cell RNA-seq. Spatial single cell RNA-seq is good tool to examine environment and location of cells. But the author only listed some technology developments, but does not describe how these technology help understanding hematopoiesis from iPSCs.

Section 4 and 5 introduce new platform to generate HSCs and the promise of PSCs-derived immune cells. Both are important research fields of PSCs. There is no single-cell related content, and thus both chapter are not related to the theme of this review.

In summary, the content may be added and reorganized. One way is to focus on Single cell RNA-seq and single cell barcoding, and summarize the published studies that discipline how to make HSCs from PSCs and hematopoiesis development.

Author Response

Point-by-point response

Reviewer 1

  1. In this review, the recent progress of PSC and advanced technologies, such as molecular barcoding, single-cell RNA-seq, spatial transcriptomics, and organoid-related platforms are summarized. The review for the application of single cell technology to understand the production of HSCs and developmental hematopoiesis from PSCs will benefit the community. The biggest problem of this review is that, there is no enough description on how “single cell” helps “hematopoietic cells from PSCs” research. The manuscript lacks the summary on the achieved findings on PSCs to hematopoiesis by single cell technology, also lacks the perspective of single-cell on PSCs research.

We thank the Reviewer for the precise and positive assessment. We appreciate the two suggestions which help dig a deep insight on deciphering hematopoiesis by single-cell technology plus molecular barcoding technique. Here are the revised parts.

P4 line 128

“To conclude, single-cell technology deciphers the heterogeneity of pre-HSCs and helps discover rare populations (CD44+ arterial endothelial cells) which could differentiate into HSCs. Considering the rarity of pre-HSCs that finally contribute to HSCs, the current bulk-level analysis (RNA-seq, ATAC-seq, and ChIP-seq) would not capture the needle in the haystack. Single-cell transcriptomics data cast deep insight into what types of factors are the potential driving force of HSC development. These highlighted transcriptional factors (Runx1, Spil1) or cell surface proteins (Notch receptors, VLA-4 integrin, and CXCR4) are conducive to the generation of HSCs, however, the identification of the deterministic factors will await. Those factors came as usual suspects hinted by prevailing knowledge from murine models and zebrafish models. We assume the unique factors in human hematopoiesis that single-cell technologies will capture. According to George Q. Daley and colleagues’ work [20], when the minimized factor pool is constructed, the first step is to knock in and overexpress these factors in PSCs in vitro. Secondly, PSCs are driven to differentiate into HSCs by morphogen/cytokine based-methods, transcription factors based strategies, or the combined approaches either in vitro or in vivo. Finally, high production of HSCs could be anticipated. The incoming findings of HSC development revealed by single-cell technology would accelerate the discovery of new factors and the application of these factors to HSCs generation (Figure 2)”

P6 line 205

“In summary, spatial transcriptomics could reveal the environment and location of cells. No doubt that some new regulatory factors can be uncovered. These hits would be the crucial guidance to the development of HSCs. We should endow the PSCs with factors by knock-in approaches and preferably induce them into HSCs. In addition, new microenvironment factors such as cytokines can be engineered in PSCs to further support HSCs generation.”

P9 line 295

“These molecular barcode strategies have unique inner principles and respective limitations. CARLIN, Macs-GESTALT, Polylox, and Transposon are evolving barcode methods whereas LARRY is a type of static barcode (Table 1). CARLIN and Macs-GESTALT, which are based on CRISPR-Cas9 editing system, may face the barcode loss problem and low detection rate of transcribed barcode RNAs. Polylox is relatively complex and costly because the PacBio Single-Molecule Real-Time (SMRT) Sequencing is needed. Transposon barcode may be labor-intensive because of a large set of PCR primers and could lead to large fragment loss or unknown genome alteration in divided cells because of random integration. Somatic mutations as a new type of barcode could be used as an indicator of non-invasive lineage tracing and fate mapping in human tissues and cells without introducing exogenous cassettes into cell genomes. Even though molecular barcode strategies are not mature enough, we predict that ongoing efforts will increase the detection rate of barcode transcripts and avoid less damage to cells. Soon, we could figure out what sets of expressed or silenced genes control the development of HSCs by the comparison of HSCs with precursor cells (pre-HSCs) by giving each progenitor cell and its progeny a unique identity simultaneously. We hope these strategies could be applied frequently and advance the understanding and production of HSCs.”

  1. Section 2 describes the schematic overview of single cell in deciphering Hematopoietic Stem and Progenitor Cells. It is good to summarize some the technical development, but the review does not describe how these development help understanding hematopoiesis from PSCs. Same situation is for spatial single cell RNA-seq. Spatial single cell RNA-seq is good tool to examine environment and location of cells. But the author only listed some technology developments, but does not describe how these technology help understanding hematopoiesis from iPSCs.

We collected recent articles to clarify how this advanced technology development helped uncover the mechanism of hematopoiesis. We described the application of these technologies in the understanding and generation of HSCs. Here are the revised parts.

P4 line 128

“To conclude, single-cell technology deciphers the heterogeneity of pre-HSCs and helps discover rare populations (CD44+ arterial endothelial cells) which could differentiate into HSCs. Considering the rarity of pre-HSCs that finally contribute to HSCs, the current bulk-level analysis (RNA-seq, ATAC-seq, and ChIP-seq) would not capture the needle in the haystack. Single-cell transcriptomics data cast deep insight into what types of factors are the potential driving force of HSC development. These highlighted transcriptional factors (Runx1, Spil1) or cell surface proteins (Notch receptors, VLA-4 integrin, and CXCR4) are conducive to the generation of HSCs, however, the identification of the deterministic factors will await. Those factors came as usual suspects hinted by prevailing knowledge from murine models and zebrafish models. We assume the unique factors in human hematopoiesis that single-cell technologies will capture. According to George Q. Daley and colleagues’ work [20], when the minimized factor pool is constructed, the first step is to knock in and overexpress these factors in PSCs in vitro. Secondly, PSCs are driven to differentiate into HSCs by morphogen/cytokine based-methods, transcription factors based strategies, or the combined approaches either in vitro or in vivo. Finally, high production of HSCs could be anticipated. The incoming findings of HSC development revealed by single-cell technology would accelerate the discovery of new factors and the application of these factors to HSCs generation (Figure 2)”

P6 line 205

“In summary, spatial transcriptomics could reveal the environment and location of cells. No doubt that some new regulatory factors can be uncovered. These hits would be the crucial guidance to the development of HSCs. We should endow the PSCs with factors by knock-in approaches and preferably induce them into HSCs. In addition, new microenvironment factors such as cytokines can be engineered in PSCs to further support HSCs generation.”

  1. Section 4 and 5 introduce new platform to generate HSCs and the promise of PSCs-derived immune cells. Both are important research fields of PSCs. There is no single-cell related content, and thus both chapter are not related to the theme of this review.

We agree that sections 1-3 are more in line with the title of this review and sections 4-5 are needed to excise.

  1. In summary, the content may be added and reorganized. One way is to focus on Single cell RNA-seq and single cell barcoding, and summarize the published studies that discipline how to make HSCs from PSCs and hematopoiesis development.

Indeed, we added more details about how single-cell technology and molecular barcoding advance the understanding of hematopoiesis. Here are the revised parts.

P4 line 128

“To conclude, single-cell technology deciphers the heterogeneity of pre-HSCs and helps discover rare populations (CD44+ arterial endothelial cells) which could differentiate into HSCs. Considering the rarity of pre-HSCs that finally contribute to HSCs, the current bulk-level analysis (RNA-seq, ATAC-seq, and ChIP-seq) would not capture the needle in the haystack. Single-cell transcriptomics data cast deep insight into what types of factors are the potential driving force of HSC development. These highlighted transcriptional factors (Runx1, Spil1) or cell surface proteins (Notch receptors, VLA-4 integrin, and CXCR4) are conducive to the generation of HSCs, however, the identification of the deterministic factors will await. Those factors came as usual suspects hinted by prevailing knowledge from murine models and zebrafish models. We assume the unique factors in human hematopoiesis that single-cell technologies will capture. According to George Q. Daley and colleagues’ work [20], when the minimized factor pool is constructed, the first step is to knock in and overexpress these factors in PSCs in vitro. Secondly, PSCs are driven to differentiate into HSCs by morphogen/cytokine based-methods, transcription factors based strategies, or the combined approaches either in vitro or in vivo. Finally, high production of HSCs could be anticipated. The incoming findings of HSC development revealed by single-cell technology would accelerate the discovery of new factors and the application of these factors to HSCs generation (Figure 2)”

P6 line 205

“In summary, spatial transcriptomics could reveal the environment and location of cells. No doubt that some new regulatory factors can be uncovered. These hits would be the crucial guidance to the development of HSCs. We should endow the PSCs with factors by knock-in approaches and preferably induce them into HSCs. In addition, new microenvironment factors such as cytokines can be engineered in PSCs to further support HSCs generation.”

P9 line 295

“These molecular barcode strategies have unique inner principles and respective limitations. CARLIN, Macs-GESTALT, Polylox, and Transposon are evolving barcode methods whereas LARRY is a type of static barcode (Table 1). CARLIN and Macs-GESTALT, which are based on CRISPR-Cas9 editing system, may face the barcode loss problem and low detection rate of transcribed barcode RNAs. Polylox is relatively complex and costly because the PacBio Single-Molecule Real-Time (SMRT) Sequencing is needed. Transposon barcode may be labor-intensive because of a large set of PCR primers and could lead to large fragment loss or unknown genome alteration in divided cells because of random integration. Somatic mutations as a new type of barcode could be used as an indicator of non-invasive lineage tracing and fate mapping in human tissues and cells without introducing exogenous cassettes into cell genomes. Even though molecular barcode strategies are not mature enough, we predict that ongoing efforts will increase the detection rate of barcode transcripts and avoid less damage to cells. Soon, we could figure out what sets of expressed or silenced genes control the development of HSCs by the comparison of HSCs with precursor cells (pre-HSCs) by giving each progenitor cell and its progeny a unique identity simultaneously. We hope these strategies could be applied frequently and advance the understanding and production of HSCs.”

Reviewer 2

This article by Xiang et al, reviews the current knowledge on single cell approaches on analyzing hematopoietic stem cells, especially during in vivo and in vitro development. The review provides extensive references about the subject and informative figures which help understand the current strategies for analyzing stem cells. However, there are some major points that the authors may consider or clarify in the manscript:

1) The review article is not clear about its exact focus. While the title of the article refers to an article about methods to analyze HSC at single cell level, the introduction diverts this focus towards methods to derive HSCs from PSCs. The authors may want to clarify their main purpose for writing this review article in both the abstract and introduction to be in line with the title.

We thank the Reviewer for the suggestion about the connection between the title and contents. We appreciate the comment thatwould help us clarify our focus and improve the readability of this review. Here are the revised parts.

P1 line 2

Title: “Single-cell approaches to deconvolute the development of HSCs”

P1 line 10

Abstract: “Here, we review the recent progress and introduce advanced technologies such as single-cell RNA-seq, spatial transcriptomics, and molecular barcoding, which may help figure out the missing information during the HSC generation. We finally discuss the unsolved questions which may be conducive to HSC production, providing a promise for HSC-based immunotherapies.”

P1 line 16

Introduction title: “Efforts on HSCs identification and generation”

2) in many parts of the article, the authors describe the findings or reports made in the past in chronological order (starting by mentioning the first author’s name and then the summary of the reference). Rather than doing so, they may want to incorporate the key content of the reference in a context cohesive manner. For example, section 1 introduced hematopoietic regulation from embryonic to adult so rather than listing the references and summary, the authors should rewrite this in a way that we understand how regulation of hematopoiesis alters in each stage of life.

Similarly, the authors list different single cell sequencing and barcoding techniques in section 3 yet do not integrate references. Reviews should aim to collectively analyze and possibly provide a critique on the current methodologies available. The authors should compare the technologies and comment on how the techniques and analysis influenced or could influence in vitro derivation of HSCs.

Some comparisons about advanced methods and their application in the understanding of HSC generation were supplemented at the end of the section. Here are the revised parts.

P4 line 128

“To conclude, single-cell technology deciphers the heterogeneity of pre-HSCs and helps discover rare populations (CD44+ arterial endothelial cells) which could differentiate into HSCs. Considering the rarity of pre-HSCs that finally contribute to HSCs, the current bulk-level analysis (RNA-seq, ATAC-seq, and ChIP-seq) would not capture the needle in the haystack. Single-cell transcriptomics data cast deep insight into what types of factors are the potential driving force of HSC development. These highlighted transcriptional factors (Runx1, Spil1) or cell surface proteins (Notch receptors, VLA-4 integrin, and CXCR4) are conducive to the generation of HSCs, however, the identification of the deterministic factors will await. Those factors came as usual suspects hinted by prevailing knowledge from murine models and zebrafish models. We assume the unique factors in human hematopoiesis that single-cell technologies will capture. According to George Q. Daley and colleagues’ work [20], when the minimized factor pool is constructed, the first step is to knock in and overexpress these factors in PSCs in vitro. Secondly, PSCs are driven to differentiate into HSCs by morphogen/cytokine based-methods, transcription factors based strategies, or the combined approaches either in vitro or in vivo. Finally, high production of HSCs could be anticipated. The incoming findings of HSC development revealed by single-cell technology would accelerate the discovery of new factors and the application of these factors to HSCs generation (Figure 2)”

P6 line 205

“In summary, spatial transcriptomics could reveal the environment and location of cells. No doubt that some new regulatory factors can be uncovered. These hits would be the crucial guidance to the development of HSCs. We should endow the PSCs with factors by knock-in approaches and preferably induce them into HSCs. In addition, new microenvironment factors such as cytokines can be engineered in PSCs to further support HSCs generation.”

P9 line 295

“These molecular barcode strategies have unique inner principles and respective limitations. CARLIN, Macs-GESTALT, Polylox, and Transposon are evolving barcode methods whereas LARRY is a type of static barcode (Table 1). CARLIN and Macs-GESTALT, which are based on CRISPR-Cas9 editing system, may face the barcode loss problem and low detection rate of transcribed barcode RNAs. Polylox is relatively complex and costly because the PacBio Single-Molecule Real-Time (SMRT) Sequencing is needed. Transposon barcode may be labor-intensive because of a large set of PCR primers and could lead to large fragment loss or unknown genome alteration in divided cells because of random integration. Somatic mutations as a new type of barcode could be used as an indicator of non-invasive lineage tracing and fate mapping in human tissues and cells without introducing exogenous cassettes into cell genomes. Even though molecular barcode strategies are not mature enough, we predict that ongoing efforts will increase the detection rate of barcode transcripts and avoid less damage to cells. Soon, we could figure out what sets of expressed or silenced genes control the development of HSCs by the comparison of HSCs with precursor cells (pre-HSCs) by giving each progenitor cell and its progeny a unique identity simultaneously. We hope these strategies could be applied frequently and advance the understanding and production of HSCs.”

3) I am not sure where the figures should be incorporated to the text. The text does not mention which figure to refer to.

We have revised and mentioned the figure within the corresponding text.  Here are the revised parts.

P2 line 60 “Later, they generated HE from hPSC in vitro, prove that the HE lineage dependent on Notch signaling is restricted to CD34+ CD43- CD184- CD73- DLL4- CD45- population (Figure 1)”

P2 line 63 “they could generate HSCs through the teratoma formation (Figure 1)”

P4 line 140 “Likely, incoming findings of HSC development revealed by single-cell technology would accelerate the discovery of new factors and the application of these factors to HSCs generation (Figure 2)”

P5 line 185 “Slide-seq worked as follows(Figure 3.a).”

P6 line 216 “Recently, molecular barcoding methods pave the way for mentioned issues(Figure 3.b).”

P9 line 297 “CARLIN, Macs-GESTALT, Polylox and Transposon are evolving barcode methods whereas LARRY is a type of static barcode (Table 1).”

4) when mentioning the methodologies, the authors provide a descriptive account of the methods for each. This should be unnecessary if the authors do not intend to compare or discuss the details of the methodologies.

Analysis, critique and comparison of these methods were added. Here are revised parts.

P9 line 295

“These molecular barcode strategies have unique inner principles and respective limitations. CARLIN, Macs-GESTALT, Polylox, and Transposon are evolving barcode methods whereas LARRY is a type of static barcode (Table 1). CARLIN and Macs-GESTALT, which are based on CRISPR-Cas9 editing system, may face the barcode loss problem and low detection rate of transcribed barcode RNAs. Polylox is relatively complex and costly because the PacBio Single-Molecule Real-Time (SMRT) Sequencing is needed. Transposon barcode may be labor-intensive because of a large set of PCR primers and could lead to large fragment loss or unknown genome alteration in divided cells because of random integration. Somatic mutations as a new type of barcode could be used as an indicator of non-invasive lineage tracing and fate mapping in human tissues and cells without introducing exogenous cassettes into cell genomes. Even though molecular barcode strategies are not mature enough, we predict that ongoing efforts will increase the detection rate of barcode transcripts and avoid less damage to cells. Soon, we could figure out what sets of expressed or silenced genes control the development of HSCs by the comparison of HSCs with precursor cells (pre-HSCs) by giving each progenitor cell and its progeny a unique identity simultaneously. We hope these strategies could be applied frequently and advance the understanding and production of HSCs.”

5) the authors should provide a conclusion containing remaining/unsolved problems and insights to future research.

We provided a third chapter “Outlook for HSCs generation” and contained the conclusions or unsolved questions about HSCs production.  Here are the revised parts.

P9 line 315

“3. Outlook for HSCs generation.

Based on our knowledge of hematopoiesis, there are several key nodes in the development of HSCs, such as endothelial cells(ECs), hemogenic endothelial cells(HECs), hematopoietic stem cells (HSCs). However, the production of HSCs is not a clear-cut linear process. From source cells to HSCs, many factors are involved. For instance, different combinations of transcription factors (TFs) and heterogenous HSC niche cell supporters suggest that there are more than one programs that drive the fate of HSCs. Until now, the question of what combinations of factors directly drive the decision to HSCs remains puzzled.[48]

By recent advanced techniques such as molecular barcoding and spatial transcriptomics combined with single-cell RNA-seq, we predict these would solve some questions. First, barcode-related lineage tracing bridges the connection between progenitor cells and progenies, not only revealing the information of major clusters but also dissecting the origins and distributions of subclones. Second, cell-cell interaction could be mapped within spatial context by spatial transcriptomics. Finally, when we continue to uncover the panorama of the development of HSCs, we could integrate these already known data and predict the best path for HSC generation based on Bayesian models. We are tantalizingly close to the state that HSCs could be produced efficiently by the guidance and multiple combinations of advanced methods from mathematics, bioinformatics, and stem cell biology.”

Minor points:

  • p2 line 72 : what is the first and second strategy?

We have revised the structure of the title and paragraph.

P2 line 69

“We here refer to two practical methods to generate HSCs in vitro and in vivo for now. 1) Directed differentiation strategies by cell-extrinsic cytokines/morphogens or 2) Direct Conversion through transcription factors (TF).”

  • P2 line 91 : still low efficient? Seems grammatically wrong

We have rewritten the sentence.

P2 line 91

“However, the HSC production in vitro or in vivo is low, which could be attributed to the lack of understanding of hematopoiesis”

  • P3 line 102: what does the morphology and markers in chaper 1 refer to?Should this be section and not chapter

We are sorry to bring vagueness. Here is the revised version.

P3 line 101

“Although the markers of HSCs and their progenitors have defined a snapshot image of each stage of HSC development.”

  • Page4 line 144 : Sub 3-1?Tense of the verbs should be revised and corrected

We have revised it.

P5 line 155-167

“Xiaowei Zhuang and colleagues developed a new spatial single-cell RNA-seq strategy termed ‘MERFISH’. For every RNA species, each had bound four encoding probes of N probes. During the sequencing process, if fluorescence was detected in one RNA species, then this RNA got a “1” bit. Each RNA with four "1" s could be identified. Researchers detected 1000 transcripts from 100 cells within 16 rounds and continued to make improvements in the detected numbers of RNA transcripts from more cells at once. Recently, Siyuan Wang and his colleagues applied Merfish to the investigation of the spatial transcriptomic profile of single cells in Ter2 -/- and WT mice fetal liver. They found that loss of Ter2 augmented the HSC population and enhanced the Wnt and Notch signaling genes in the HSC niche. They also validated the HSCs directly inter-acted with endothelial cells (ECs), and two subtypes of ECs mainly made different contributions to the HSC niche by their unique signaling molecules. Finally, hepatocytes and megakaryocytes were discovered to support HSC niches”

  • P10 line 321: immune cells meet the immunotherapy : expression is rather odd

Considering the theme of this review, we have excised the sections 4-5.

Reviewer 2 Report

This article by Xiang et al, reviews the current knowledge on single cell approaches on analyzing hematopoietic stem cells, especially during in vivo and in vitro development. The review provides extensive references about the subject and informative figures which help understand the current strategies for analyzing stem cells. However, there are some major points that the authors may consider or clarify in the manscript:

1) The review article is not clear about its exact focus. While the title of the article refers to an article about methods to analyze HSC at single cell level, the introduction diverts this focus towards methods to derive HSCs from PSCs. The authors may want to clarify their main purpose for writing this review article in both the abstract and introduction to be in line with the title.

2) in many parts of the article, the authors describe the findings or reports made in the past in chronological order (starting by mentioning the first author’s name and then the summary of the reference). Rather than doing so, they may want to incorporate the key content of the reference in a context cohesive manner. For example, section 1 introduced hematopoietic regulation from embryonic to adult so rather than listing the references and summary, the authors should rewrite this in a way that we understand how regulation of hematopoiesis alters in each stage of life.

Similarly, the authors list different single cell sequencing and barcoding techniques in section 3 yet do not integrate references. Reviews should aim to collectively analyze and possibly provide a critique on the current methodologies available. The authors should compare the technologies and comment on how the techniques and analysis influenced or could influence in vitro derivation of HSCs.

3) I am not sure where the figures should be incorporated to the text. The text does not mention which figure to refer to.

4) when mentioning the methodologies, theauthors provide a descriptive account of the methods for each. This should be unnecessary if the authors do not intend to compare or discuss the details of the methodologies.

5) the authors should provide a conclusion containing remaining/unsolved problems and insights to future research.

Minor points:

  • p2 line 72 : what is the first and second strategy?
  • P2 line 91 : still low efficient? Seems grammatically wrong
  • P3 line 102: what does the morphology and markers in chaper 1 refer to?
  • Should this be section and not chapter
  • Page4 line 144 : Sub 3-1?
  • Tense of the verbs should be revised and corrected
  • P10 line 321: immune cells meet the immunotherapy : expression is rather odd

Author Response

(The authors gave the same response as above.)

Round 2

Reviewer 1 Report

 I have no further comments.

Reviewer 2 Report

The authors have answered all the concerns that were mentioned in the previous review. I have no further concerns or comments.